# CD4-Positive T-Cell Responses to MOG Peptides in MOG Antibody-Associated Disease

**DOI:** 10.3390/ijms26083606

**Published:** 2025-04-11

**Authors:** Hirohiko Ono, Tatsuro Misu, Chihiro Namatame, Yuki Matsumoto, Yoshiki Takai, Shuhei Nishiyama, Hiroshi Kuroda, Toshiyuki Takahashi, Ichiro Nakashima, Kazuo Fujihara, Masashi Aoki

**Affiliations:** 1Department of Neurology, Tohoku University Graduate School of Medicine, Sendai 980-0845, Japan; hirohiko.ono.b2@tohoku.ac.jp (H.O.); chihiro.namatame.e6@tohoku.ac.jp (C.N.); yuki.matsumoto.e5@tohoku.ac.jp (Y.M.); yoshiki.takai.e6@tohoku.ac.jp (Y.T.); shuhei.nishiyama.b7@tohoku.ac.jp (S.N.); dakuro@med.tohoku.ac.jp (H.K.); masashi.aoki.c8@tohoku.ac.jp (M.A.); 2Department of Neurology, National Hospital Organization Yonezawa National Hospital, Yonezawa 992-1202, Japan; t-toshiyuki@mta.biglobe.ne.jp; 3Division of Neurology, Tohoku Medical and Pharmaceutical University, Sendai 983-8512, Japan; nakashima@tohoku-mpu.ac.jp; 4Department of Multiple Sclerosis Therapeutics, Fukushima Medical University, Fukushima 960-1295, Japan

**Keywords:** myelin oligodendrocyte glycoprotein, T-cell epitope, aquaporin-4, granulocyte–macrophage colony-stimulating factor, multiple sclerosis, neuromyelitis optica

## Abstract

To clarify T-cell responses in myelin oligodendrocyte glycoprotein (MOG) antibody-associated disease (MOGAD), we cultured the peripheral blood mononuclear cells of 24 patients with MOGAD and 20 with aquaporin-4 (AQP4) antibody-positive neuromyelitis optica spectrum disorders (NMOSD), and those of 17 healthy controls (HCs), in the presence of fourteen MOG peptides covering the full-length MOG, five AQP4 peptides, two myelin basic protein peptides, or two proteolipid protein peptides. Then, we measured T-cell activation markers, such as cell surface CD69 and the intracellular production of granulocyte–macrophage colony-stimulating factor (GM-CSF) and interferon-γ in CD4-positive T-cells, with a flow cytometer. The expression of CD69 in response to MOG p16–40 and MOG p181–205 was significantly higher (Stimulation Index > 2) in MOGAD than in HCs. Also, CD69 for AQP4 p21–40, AQP4 p211–230, and MOG p166–190 were significantly increased in NMOSD than in HCs. Intracellular GM-CSF production responding to MOG p16–40 was significantly higher in MOGAD than in HCs (*p* < 0.05), although intracellular interferon-γ was not elevated. None of the responses to the other peptides were different between the groups. The present study showed subtle CD4-positive T-cell activation elicited by some MOG peptides alone in patients with MOGAD. Further studies of cytokines or other stimulation and alternative assay markers and metrics are needed to delineate the immunopathological roles of T-cells in MOGAD.

## 1. Introduction

Myelin oligodendrocyte glycoprotein (MOG), a myelin protein expressed in the outermost layer of central nervous system (CNS)’s myelins and oligodendrocytes, is a member of the immunoglobulin (Ig) superfamily and comprises 218 amino acids, one extracellular domain, one transmembrane domain, one cytoplasmic domain, and one membrane-associated domain [1]. Although the function of MOG remains unclear, with potential links to the compaction and maintenance of myelin and cell adhesion, it has been widely used as an immunogen in experimental autoimmune encephalomyelitis (EAE), and has long been pursued in human demyelinating diseases including multiple sclerosis (MS) [2]. However, anti-MOG antibodies (MOG-ab) are rarely detected in patients with typical MS using assays with human MOG-transfected cells [3]. Conformation-sensitive MOG-ab detected by cell-based assays are positive in some cases with inflammatory CNS diseases including optic neuritis, myelitis, acute disseminated encephalomyelitis (ADEM), aquaporin-4 (AQP4) antibody (AQP4-ab)-negative neuromyelitis optica spectrum disorder (NMOSD), brainstem encephalitis, and cerebral cortical encephalitis, which are now called as MOG-ab-associated disease (MOGAD) [4].

MOG-ab are pathogenic in vitro [5] and MOG-ab, mainly IgG1 [3], require T-cells for their generation, indicating the pathogenetic importance of T-cells in MOGAD. Some patients with MOGAD have experienced relapses after reverting to seronegativity [6], in which cellular immunity to MOG may play a pivotal role. Moreover, lymphocytes infiltrating in the perivascular regions in the acute phase lesions of MOGAD are mainly CD4-positive T-cells [7]. However, the features of T-cells in the pathogenesis of MOGAD are poorly understood [8]. In the present study, we aimed to identify MOG-reactive CD4-positive T-cells in MOGAD.

## 2. Results

### 2.1. Clinical Demographics of Study Participants

The clinical features of the 24 patients with MOGAD (12 females and 12 males), 20 patients with AQP4-ab-positive NMOSD (hereinafter called NMOSD) (19 females and 1 male), and 17 HCs (14 females and 3 males) enrolled in the study are shown in Table 1. The mean ages at blood collection were 42.1 ± 13.2 and 57.0 ± 12.1 years in the MOGAD and NMOSD groups, respectively (*p* = 0.001). The disease durations were 5.1 (0.1–27) and 12.2 (0.1–25) years in the MOGAD and NMOSD groups, respectively (*p* = 0.005). In the MOGAD group, the clinical diagnoses were ADEM (*n* = 7), optic neuritis (*n* = 13), myelitis (*n* = 7), brainstem encephalitis (*n* = 5), cerebral cortical encephalitis (*n* = 2), and multiple brain lesions without encephalopathy (*n* = 7). There was no MOGAD patient who met the international consensus criteria for NMOSD without AQP4-IgG. While 11 MOGAD patients (45%) experienced relapses, with a median of 2.1 attacks, 12 NMOSD patients (60%) had a significantly higher number of relapses, with a median of 2.7 attacks (*p* = 0.35). Three of the eleven MOGAD (45%) patients treated with oral steroids at the time of blood sampling were also receiving azathioprine (AZA). One MOGAD patient was receiving methotrexate (MTX) monotherapy due to rheumatoid arthritis, and another MOGAD patient was receiving mesalazine monotherapy due to ulcerative colitis. Nearly all NMOSD patients (19/20) were treated with oral prednisolone (5 mg/day or less).

### 2.2. CD4-Positive T-Cells of MOGAD Patients Respond to MOG p16–40 and MOG p181–205

First, we determined T-cell responses to MOG peptides by measuring the expression levels of CD69, an early-phase T-cell activation marker. The representative gating strategy for T-cell response to MOG p16–40 is shown in Figure 1A. The percentage of CD69 positives in cultured CD4-positive T-cells increased from 8.6% in the absence of MOG p16–40 to 20.2% in the presence of MOG p16–40. The SI was 2.3 in culture incubated with MOG p16–40. We compared the unstimulated state in three groups to see if there was a difference in the CD69 expression of the T-cells in no-peptide conditions, but no significant difference was observed. T-cell activation, defined as an SI > 2 in response to at least one MOG peptide, was observed in 42% (10/24) of the MOGAD patients, 45% (9/20) of the NMOSD patients, and 29% (5/17) of the HCs. Moreover, the T-cell responses to MOG p16–40 and MOG p181–205 were significantly stronger in the MOGAD patients than in the HCs (adjusted *p*, 0.02 and <0.0001, respectively; Figure 1B). An SI > 2 in response to at least one AQP4 peptide was observed in 21% (5/24) of the MOGAD patients, 20% (4/20) of the NMOSD patients, and 0% (0/17) of the HCs. T-cell responses to AQP4 p21–40, AQP4 p211–230, and MOG p166–190 were significantly stronger in the NMOSD patients than in the HCs (adjusted *p*, 0.03, 0.04, and 0.008, respectively; Figure 1B). The T-cells of two MOGAD patients and one NMOSD patient responded to MBP, whereas the T-cells of one MOGAD patient and two NMOSD patients responded to PLP. In contrast, the T-cells of the HCs did not respond to MBP or PLP. There was no significant difference in SI between the MOGAD patients and NMOSD patients.

When comparing the treated and untreated MOGAD patients, no difference was observed in the expression of CD69 on T-cells between the two groups. Regarding the impact of relapses on antigen response and epitope spreading, we compared the SI values between the patients with one attack and those with two or more attacks. In both the MOG-ab-positive and AQP4-ab-positive patient groups, there was no significant difference in the SI between the monophasic patients and those with recurrent episodes (Figure 1C), suggesting that epitope spreading had a minimal effect on T-cell responses in this experiment.

### 2.3. CD4+ T-Cells of MOGAD Patients Produced GM-CSF in Response to MOG p16–40

Next, we tested the ability of T-cells to produce inflammatory cytokines in response to CNS peptides. A representative gating of T-cell intracellular cytokine production to MOG peptides in a patient with MOGAD (43-year-old male, treated with PSL and AZA) is shown in Figure 2A. Granulocyte–macrophage colony-stimulating factor (GM-CSF) production by T-cells, defined as an SI > 2 in response to at least one MOG peptide, was observed in 100% (24/24) of the MOGAD patients, 65% (13/20) of the NMOSD patients, and 35% (6/17) of the HCs. Moreover, the GM-CSF production following exposure to MOG p16–40 was significantly increased in the T-cells of MOGAD patients than in those of the HCs (adjusted *p*, 0.01; Figure 2B). In contrast, the GM-CSF production following CNS peptide exposure was not significantly different between the T-cells of NMOSD patients and those of the HCs. Interferon (IFN)-γ production from the T-cells in response to at least one MOG peptide was observed in 54% (13/24) of the MOGAD patients, 55% (11/20) of the NMOSD patients, and 29% (5/17) of the HCs, although no significant IFN-γ production was observed in the MOGAD and NMOSD patients (Figure 2C).

### 2.4. Relationship Between Disease Activity and T-Cell Response to MOG Peptides in Three Patients with MOGAD

We next evaluated the relationship between disease activity and T-cell response to CNS peptides in three MOGAD patients (Figure 3), as summarized below.

A 52-year-old male patient with a previous history of encephalopathy with left optic neuritis that occurred 20 years prior visited our hospital for a regular follow-up, at which time he had no complaints; the first blood sample was collected on that day. One month later, he developed blurred vision in his right eye and was admitted to our hospital, which was when the second blood sample was collected. His vision improved soon after the initiation of high-dose intravenous methylprednisolone (IVMP) treatment, and the third blood sample was collected after IVMP. An increase in SI to >2 was observed in T-cells responding to MOG p16–40 (SI = 2.02), MOG p166–190 (SI = 3.05), and MOG p181–205 (SI = 2.97) (Figure 3A).

A 52-year-old male with no history of neurological diseases developed acute paraparesis and bladder and bowel dysfunctions and was admitted to our hospital. Magnetic resonance imaging (MRI) revealed multiple brainstem and spinal lesions with gadolinium enhancement, and the first blood sample was collected on the day of the MRI studies. After repeated IVMP treatment, his paraparesis improved and the lesions with gadolinium enhancement disappeared. The second and third blood samples were collected after the second and third courses of IVMP treatment, respectively. An increase in SI to >2 was observed in the T-cells responding to MOG p1–25 (SI = 2.69), MOG p16–40 (SI = 2.53), MOG p136–160 (SI = 2.39), MOG p181–205 (SI = 3.96), MOG p196–218 (SI = 2.55), AQP4 p21–40 (SI = 2.38), and AQP4 p211–230 (SI = 2.33) (Figure 3B).

A 49-year-old female patient with four optic neuritis episodes who received oral prednisolone (13 mg/day) and azathioprine (50 mg/day) developed left ocular pain and blurred vision and was admitted to our hospital, and the first blood sample was collected on the day of admission. Her vision deteriorated to no light perception within seven days; the second blood sample was collected at that time. She was then treated with IVMP and plasma exchange, and her vision gradually improved within one month, when the third blood sample was collected. An increase in SI to >2 was observed in the T-cells responding to MOG p106–130 (SI = 2.07) and AQP4 p21–40 (SI = 2.04) (Figure 3C).

In all three patients, higher SIs to MOG or other CNS peptides were observed in the acute phase of MOGAD.

## 3. Discussion

In the present study, we showed that freshly isolated T-cells from the peripheral blood samples of MOG-ab+-positive patients exhibited increased responses to MOG p16–40 and MOG p181–205, and interestingly, GM-CSF production was increased in response to MOG p16–40, which is localized in the extracellular Ig-like domain of MOG. On the other hand, in NMOSD, we found elevated T-cell responses to AQP4 p21–40, AQP4 p211–230, and MOG p166–190 in NMOSD patients, which is in partial agreement with previous studies investigating AQP4-reactive T-cell responses in NMOSD (increased T-cell responses to AQP4 p11–30, p21–40, p61–80, p131–150, p156–170, p211–230, and p261–280 [SI > 2] were seen in NMOSD) [9,10].

Several immunodominant MOG epitopes have been reported in MS, including the Ig-like extracellular domain as well as the transmembrane and intracellular domains [11,12]. MOG, located 100 kb telomeric to human leukocyte antigen (HLA) class I (*HLA F*), is highly immunogenic and has been suspected to be an important target of demyelinating diseases such as MS [13]. The reported T-cell epitopes in MOG which are associated with increased susceptibility to MS include MOG p146–154 [14], MOG p1–120 [15], MOG p119–132, MOG p181–195, and p186–200 [16]; some of these results were similar to our results on the transmembrane MOG p181–205 epitope in MOGAD and MOG p166–190 in NMOSD. However, there is no consensus on the identification of common MOG epitopes of auto-reactive T-cells and the MOG-ab in demyelinating diseases such as MS, NMOSD, or MOGAD by enzyme-linked immunosorbent assays [17] or by cell-based assays. Sun et al. reported that pediatric-onset patients with MOGAD were associated with the HLA DQB1*05:02-DRB1*16:02 allele [18]. Using a bioinformatic analysis, they identified a strong epitope located in the extracellular domain of MOG p37–45 binding to DQB1*05:02; this is the epitope which partially overlaps with MOG p16–40 which is identified in the present study. In a recent study, Hofer et al. investigated T-cell responses to nine MOG peptides and eight AQP4 peptides in MOGAD and NMOSD patients using the carboxyfluorescein diacetate succinimidyl ester (CSFE) proliferation assay [19], and they identified the T-cell epitope in AQP4 p156–170 in AQP4-ab-positive patients but did not find any T-cell epitopes in MOG in patients with MOGAD; however, they did not evaluate the full-length MOG including MOG p16–40. Another possible explanation is the low number of participants in the previous study, which only included ten MOGAD patients and eight NMOSD patients.

GM-CSF is a cytokine that promotes the differentiation, proliferation, and activation of monocytes and dendritic cells in autoimmune inflammatory diseases such as MS and rheumatoid arthritis [20]. In animal studies, GM-CSF-deficient mice are resistant to EAE [21], and GM-CSF produced by auto-reactive T-cells has been shown to promote microglial activation and EAE onset [22]. Activation of the microglia may facilitate local antigen presentation and the production of pro-inflammatory cytokines in the CNS, which may lead to inflammatory demyelination. Meanwhile, pro-inflammatory cytokines, including tumor necrosis factor (TNF)-α and interleukin (IL)-1β secreted by activated microglia, can increase the permeability of the blood–brain barrier (BBB) [20], and our previous study demonstrated the increased levels of pro-inflammatory cytokines such as GM-CSF, TNF-α, IL-6, and IFN-γ in the CSF during attacks in patients with MOGAD compared with controls [23]. In MS, pathogenic CD4+ T-cells produce both IFN-γ and GM-CSF [9]. On the other hand, in the present study of MOGAD, CD4-positive T-cells stimulated by MOG p16–40 produced GM-CSF but did not produce IFN-γ. This feature of the skewed production of pro-inflammatory cytokines by T-cells in MOGAD may be similar to the T-cells which only produce GM-CSF, called “ThG cells” [9]. Although ThG cells are also reported to be increased in MS [9], the immunopathological role is not well elucidated. Thus, it is intriguing to further analyze whether the GM-CSF production by CD4-positive T-cells detected in the present study of MOGAD was due to “ThG cells”.

Recently, we reported that ADEM-like perivenous demyelination with MOG-dominant myelin loss was a characteristic pathological feature of MOGAD. In the acute stage of MOGAD, we also observed large numbers of perivascular infiltrated lymphocytes with CD4-positive T-cell dominance, which is distinct from the CD8-positive T-cell dominance in MS. But perivascular deposits of activated complements and Ig were much less remarkable in MOGAD than in NMOSD [7]. Therefore, the pathogenetic roles of T-cells and B-cells appear to be different in MOGAD and NMOSD.

Lassmann et al. reported the influence of the balance between T-cells and MOG antibodies on the characteristics of inflammatory demyelinating lesions in EAE [24]. In EAE induced by a higher number of T-cells, the perivascular inflammatory infiltrates were observed throughout the entire medulla oblongata and spinal cord, surrounded by sleeves of demyelination like ADEM. In contrast, in EAE induced by a lower number of T-cells, larger and confluent, albeit sparce, demyelinated lesions were observed in areas near the CSF space, which is similar to the MS pathology. Moreover, the intravenous injection of MOG-ab alone did not induce demyelinating lesions, inflammation, or clinical symptoms. These findings indicate that the synergistic action of T-cells and autoantibodies is necessary for the induction of inflammatory CNS demyelination and that their quantities impact the lesion volume and distribution. In more recent studies, Bettelli et al. and Krishnamoorthy et al. reported a double transgenic mouse spontaneously exaggerated model of MOG-EAE, which revealed the interplay between MOG-specific T-cell receptors as well as B-cells in producing antibodies to MOG for the induction of EAE [25,26].

In a previous study conducted by Winklemeier et al., B-cells derived from the peripheral blood of about 60% of MOGAD patients produced MOG-ab in vitro. Those MOG-ab detected by a live cell-based assay mainly bound to the extracellular domain of MOGs [27]. They also reported that MOG-specific B-cells in the blood showed no correlation with serum MOG-ab levels and the epitopes recognized by the MOG-ab were intra-individually heterogeneous [27]. In addition, Tea et al. revealed that in more than 75% of pediatric and adult cases of MOGAD, the MOG-ab bound to a dominant extracellular antigenic region around Proline42 [28] (which is close to MOG p16–40, which we detected in the present study of CD4-positive T-cells).

The present study has several limitations that should be acknowledged. First, this was a single-center study in Japanese patients with MOGAD, and thus our findings should be confirmed in larger scale studies of pediatric and adult patients with different ethnic backgrounds. Second, about half of the patients with MOGAD were receiving immunosuppressive treatment and nearly all patients with NMOSD were on oral prednisolone treatment. These treatments might have been associated with the relatively low T-cell responses detected in the present study. Third, we focused on early T-cell activation with MOG peptides; however, we did not analyze late T-cell activation markers such as CD154, nor did we conduct a T-cell proliferation using the CSFE proliferation assay or [3H] the thymidine uptake assay. SI > 2 was defined as significant in our study, as in the previous report [10]; however, SI > 3 was considered to be significant in some other studies [9,19,29]. Additionally, we did not observe a strong T-cell response elicited by MOG peptides alone in patients with MOGAD. The reason why we only observed subtle T-cell responses to MOG peptides can be explained by the fact that extra stimulations, including IL-2 [29] and IL-7 [30], may be necessary for the induction of a strong T-cell response. The inflammatory condition in MOGAD could not be reproduced in our experimental method. Moreover, the limited evaluation of intracellular cytokines (IL-4, IL-17, IL-10, etc.) and the lack of HLA phenotyping were also limitations of this study. Adding those data would be useful to further clarify T-cell responses to MOG peptides in MOGAD.

## 4. Methods

### 4.1. Selection of Patients and Controls

In addition to the 24 adult patients with MOGAD [31] and the 20 with NMOSD [32], 17 adult healthy controls (HCs) recruited from hospital volunteers were included in the present study (Table 1). Data were presented as numbers, means ± standard deviation, or medians (range). All participants were tested for MOG-ab and AQP4-ab using our previously reported in-house cell-based assay [33]. To avoid the influence of treatment with oral steroids or immunosuppressants on T-cell function, this study included NMOSD patients on an oral steroid treatment of <5 mg/day without any immunosuppressive drugs. The median oral steroid doses were 0 and 5.0 mg in the MOGAD and NMOSD patients, respectively.

### 4.2. Standard Protocol Approval, Registration, and Patient Consent

This study was approved by the Ethics Committee of Tohoku University School of Medicine, and written informed consent was obtained from all participants.

### 4.3. Peptides

As antigens, 14 MOG peptides within overlapping regions of full-length MOG with a purity > 90% were commercially synthesized by GenScript (Piscataway, NJ, USA). Each peptide ranged from 21 to 25 amino acids in length and overlapped by 10 amino acids. In addition, five AQP4, two proteolipid protein (PLP), and two myelin basic protein (MBP) peptides, which were previously reported to be identified [9,11,34], were commercially synthesized, with purities > 90% (GenScript). The sequences of the synthesized peptides are listed in Table 2.

### 4.4. Cell Preparation and Culture

PBMCs were isolated by density gradient centrifugation using Ficoll-Paque PLUS (GE Healthcare Bioscience, Uppsala, Sweden). All blood samples were tested for MOG-ab and AQP4-ab at the time of flow cytometric analysis using the in-house cell-based assay. For the T-cell activation assay, freshly isolated PBMCs were seeded in 96-well flat-bottom plates at a density of 5 × 10^5^ cells/well in AIM-V medium (Life Technologies, Grand Island, New York, NY, USA) and cultured for two days in medium alone. Next, specific synthesized peptides were added to the wells at a final concentration of 10 μg/mL and PBMCs were incubated with the antigens for four hours [9]. For the intracellular cytokine assay of T-cells, freshly isolated PBMCs were seeded in 96-well flat-bottom plates at a density of 5 × 10^5^ cells/well in AIM-V medium and cultured with specific synthesized peptides at a final concentration of 10 μg/mL for seven days [19].

### 4.5. Flow Cytometry Analysis

For the T-cell activation assay, incubated PBMCs were harvested, washed, and stained with anti-CD3-APC (BioLegend, San Diego, CA, USA), anti-CD4-PerCP-Cy5.5 (BioLegend), and anti-CD69-FITC (BioLegend) antibodies. After incubation for 30 min, the PBMCs were washed and resuspended in phosphate-buffered saline supplemented with 0.5% bovine serum albumin and analyzed by FACS Verse (BD Biosciences, Franklin Lakes, NJ, USA), according to the manufacturer’s instructions. For the analysis of the antigenic T-cell response, the stimulation index (SI) was calculated as follows, and an SI > 2 was considered as significant [10]:$$SI=\frac{CD69-positive\;CD4\;T\;cells\;with\;one\;peptide\;\left(\%\right)}{CD69-positive\;CD4\;T\;cells\;without\;peptide\;\left(medium\;alone\right)\;\left(\%\right)\;}$$

For the T-cell intracellular cytokine assay, GolgiStop, a membrane protein carrier blocker, was added four hours before harvesting the PBMCs that were incubated for seven days. Next, the cells were washed and stained with anti-CD3-APC-Cy7 and anti-CD4-PerCP-Cy5.5 antibodies, followed by fixation and permeabilization using the Cytofix/Cytoperm Plus fixation/permeabilization kit (BD Biosciences). The cells were then stained with anti-IFN-γ-FITC and anti-GM-CSF-PE antibodies. Flow cytometric analysis and SI calculations were performed as described above.

### 4.6. Statistical Analysis of Flow Cytometry Data

The FlowJo software Version 10.4.2 (Tree Star, Ashland, OR, USA) was used for flow cytometric data analysis. All statistical analyses were performed using GraphPad Prism 6 (GraphPad Software, La Jolla, CA, USA). The Mann–Whitney *U* test was used to compare disease duration and the number of attacks between groups. A two-way analysis of variance with Bonferroni’s multiple comparisons test was used to compare the SIs between the three groups. An adjusted *p* < 0.05 was considered to indicate statistical significance.

## 5. Conclusions

In the present study, subtle CD4-positive T-cell activation elicited by some MOG peptides alone was observed in patients with MOGAD. Further studies with cytokines or other stimulation and alternative assay markers (lymphocyte proliferation assay, etc.) and metrics (SI > 3, etc.) are needed to delineate the immunopathological roles of T-cells in MOGAD.

## Figures and Tables

**Figure 1 ijms-26-03606-f001:**
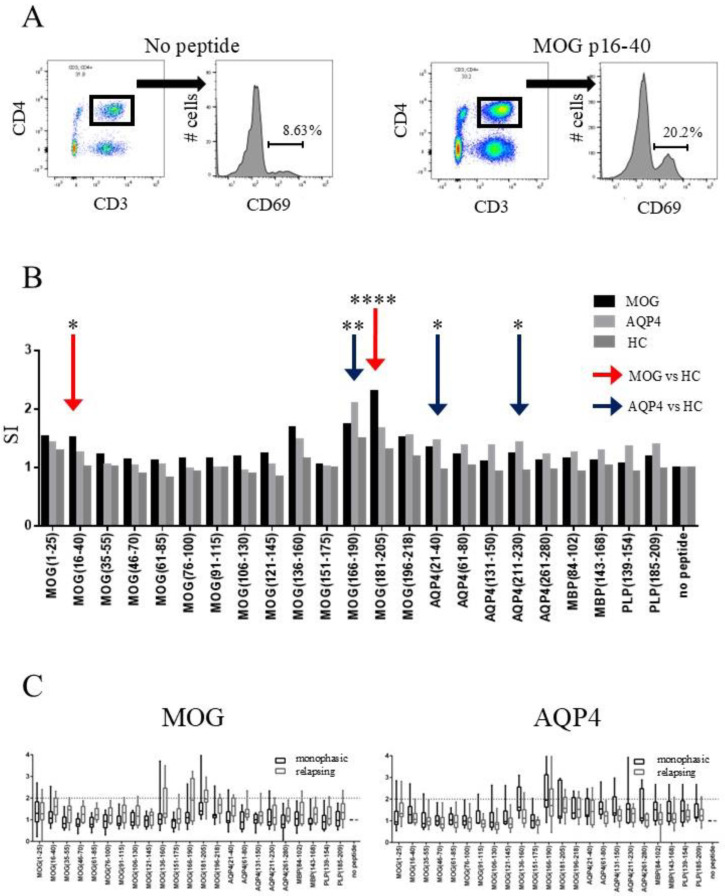
Expression of CD69 in CD4-positive T-cells in response to MOG, AQP4, and other myelin peptides. (**A**) Representative gating of T-cell response to MOG peptides in patient with MOGAD. (**B**) Comparison of T-cell responses to MOG and AQP4 peptides among MOGAD and NMOSD patients and HCs. Bars show mean SI values. Red arrows indicate peptides with significantly higher SI values in MOGAD patients compared with HCs. Blue arrows indicate peptides with significantly higher SI values in NMOSD patients compared with HCs. (**C**) Comparison of SI values between patients with monophasic and relapsing diseases. * *p* < 0.05, ** *p* < 0.01, **** *p* < 0.0001. MOG = myelin oligodendrocyte glycoprotein; SI = stimulation index; AQP4 = aquaporin-4; HCs = healthy controls.

**Figure 2 ijms-26-03606-f002:**
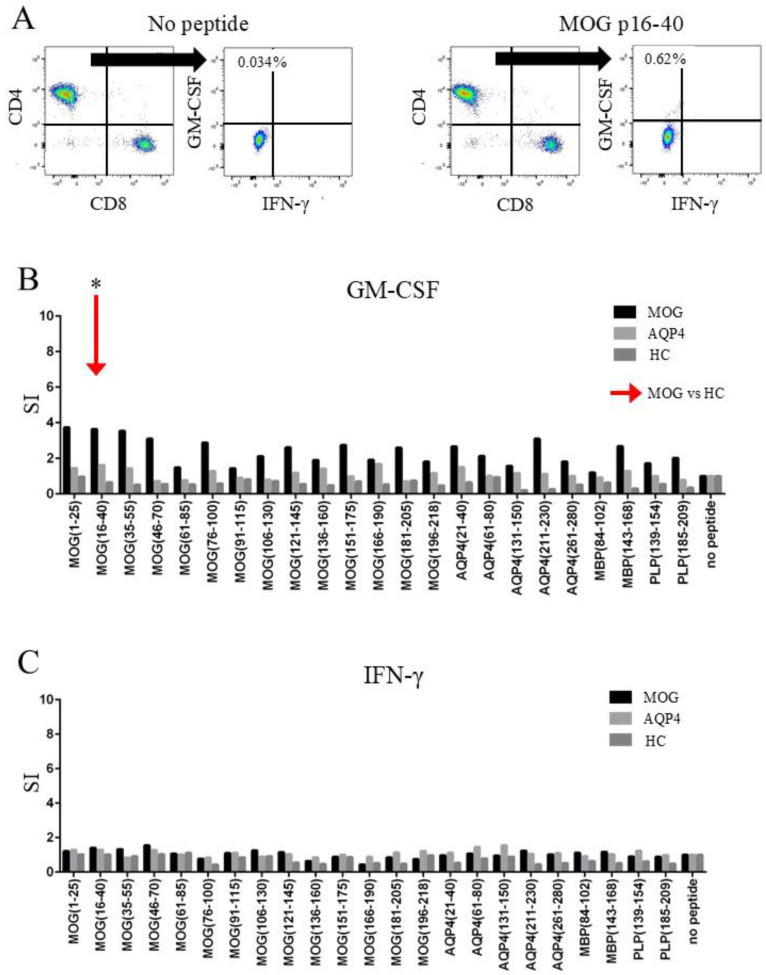
The CD4-positive T-cells of MOGAD patients produced GM-CSF in response to MOG p16–40. (**A**) A representative gating of T-cell intracellular cytokine production to a MOG peptide in a patient with MOGAD. Intracellular GM-CSF in CD4-positive cells was increased from 0.034% (with no peptide) to 0.62% (with MOGp16–40). Intracellular levels of GM-CSF (**B**) and IFN-γ (**C**) in CD4-positive T-cells were determined after stimulation with specific peptides. Note the significant increase in SI in intracellular GM-CSF in response to MOG p16–40 in the MOGAD patients compared with the HCs (red arrow). * *p* < 0.05. SI = stimulation index; MOG = myelin oligodendrocyte glycoprotein; AQP4 = aquaporin-4; HC = healthy control; MBP = myelin basic protein; PLP = proteolipid protein; GM-CSF = granulocyte macrophage colony-stimulating factor; IFN-γ = interferon-γ.

**Figure 3 ijms-26-03606-f003:**
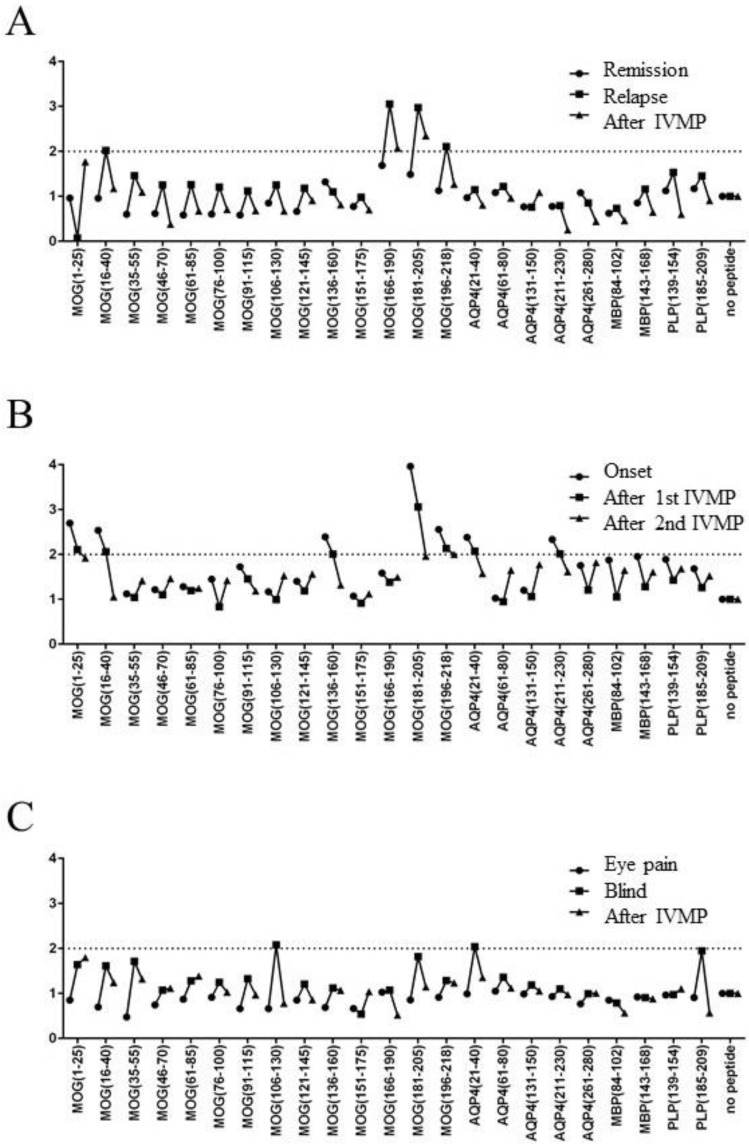
T-cell responses to MOG and other peptides before, during, and after attacks in three patients with MOGAD; T-cell responses to several peptides using the CD69 activation assay were evaluated before, during, and after clinical relapse in three MOGAD patients. (**A**) A 52-year-old patient with optic neuritis, (**B**) a 52-year-old male patient with myelitis, and (**C**) a 49-year-old female patient with optic neuritis. The SIs of dotted lines are 2.00. IVMP = intravenous methylprednisolone; MOG = myelin oligodendrocyte glycoprotein; AQP4 = aquaporin-4; MBP = myelin basic protein; PLP = proteolipid protein; SI = stimulation index.

**Table 1 ijms-26-03606-t001:** Demographic and clinical features of participants.

	MOGAD (*n* = 24)	NMOSD (*n* = 20)	HC (*n* = 17)
**sex ratio (male/female)**	12:12	1:19	3:14
**age, years (mean ± SD)**	42.1 ± 13.2	57.0 ± 12.1	33.8 ± 4.4
**disease duration, year, median (range)**	5.1 (0.1–27)	12.2 (0.1–25)	**-**
**number of attacks, median (range)**	2.1 (1–5)	2.7 (1–13)	**-**
**time interval between blood sampling and last attack, year, median (range)**	1.0 (0–14)	6.5 (1–14)	**-**
**number of patients under oral steroid treatment**	11	19	**-**
**dose of oral steroid, mg/day, median**	0	5.0	**-**
**immunosuppressive treatment**	AZA (*n* = 3), MTX (*n* = 1), mesalazine (*n* = 1)	AZA (*n* = 1)	-

Abbreviations: MOGAD = anti-myelin oligodendrocyte glycoprotein antibody associated disease; NMOSD = neuromyelitis optica spectrum disorder; HC = healthy controls; SD = standard deviation; AZA = azathioprine; MTX = methotrexate.

**Table 2 ijms-26-03606-t002:** Amino acid sequences of synthesized MOG, AQP4, PLP and MBP peptides.

	Peptides	Sequence	AA Length
**MOG**	MOG (1–25)	GQFRVIGPRHPIRALVGDEVELPCR	25
	MOG (16–40)	VGDEVELPCRISPGKNATGMEVGWY	25
	MOG (35–55)	MEVGWYRPPFSRVVHLYRNGK	21
	MOG (46–70)	RVVHLYRNGKDQDGDQAPEYRGRTE	25
	MOG (61–85)	QAPEYRGRTELLKDAIGEGKVTLRI	25
	MOG (76–100)	IGEGKVTLRIRNVRFSDEGGFTCFF	25
	MOG (91–115)	SDEGGFTCFFRDHSYQEEAAMELKV	25
	MOG (106–130)	QEEAAMELKVEDPFYWVSPGVLVLL	25
	MOG (121–145)	WVSPGVLVLLAVLPVLLLQITVGLV	25
	MOG (136–160)	LLLQITVGLVFLCLQYRLRGKLRAE	25
	MOG (151–175)	YRLRGKLRAEIENLHRTFDPHFLRV	25
	MOG (166–190)	RTFDPHFLRVPCWKITLFVIVPVLG	25
	MOG (181–205)	TLFVIVPVLGPLVALIICYNWLHRR	25
	MOG (196–218)	IICYNWLHRRLAGQFLEELRNPF	23
**AQP4**	AQP4 (21–40)	NIMVAFKGVWTQAFWKAVTA	20
	AQP4 (61–80)	GTEKPLPVDMVLISLCFGLS	20
	AQP4 (131–150)	GILYLVTPPSVVGGLGVTMV	20
	AQP4 (211–230)	SMNPARSFGPAVIMGNWENH	20
	AQP4 (261–280)	RFKEAFSKAAQQTKGSYMEV	20
**PLP**	PLP (139–154)	HCLGKWLGHPDKFVGI	16
	PLP (185–209)	SIAFPSKTSASIGSLCADARMYGVL	25
**MBP**	MBP (84–102)	DENPVVHFFKNIVTPRTPP	19
	MBP (143–168)	FKGVDAQGTLSKIFKLGGRD	20

Abbreviations: AA = amino acid; MOG = myelin oligodendrocyte glycoprotein; AQP4 = aquaporin 4; PLP = proteolipid protein; MBP = myelin basic protein.

## Data Availability

Anonymized data will be shared on request by qualified investigators.

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
