# Peer review of "CD4-Positive T-Cell Responses to MOG Peptides in MOG Antibody-Associated Disease"

_ijms, 2025, doi:10.3390/ijms26083606_

Round 1
Reviewer 1 Report
Comments and Suggestions for Authors
The authors describe results from ex vivo stimulation of peripheral blood mononuclear cells from subjects with MOGAD, NMOSD and healthy controls to test CD4+ T cell response to MOG peptides and control peptides. The primary outcome measure was the expression of CD69, an immunophenotypic marker of activation. Two MOG peptides (MOGp16-40 and MOGp181-205) were shown to significantly induce CD4+ T cell activation as indicated by CD69 expression after a short (30 min) culture. MOGp16-40 was also shown to induce significant increase in GM-CSF, but not interferon-gamma, production after a long (7 days) culture. Defining the T cell response to MOG in MOGAD, including the identification of an immunodominant peptide and the Th lineage differentiation of anti-MOG T cell response would be of high interest to the field. Due to limitations in the study design, the results of this study are preliminary rather than conclusive.
The strengths of the study are 1) the use of human samples, 2) use of a peptide library that covers the full-length MOG protein and 3) the use of appropriate subject and peptide controls. Limitations of the study are appropriately stated in the Discussion and the stated conclusions are appropriate for the study design/results.
Weaknesses of the study are 1) the lack of a measure of proliferation, 2) limited assessment of intracellular cytokines (i.e. lacks assessment of IL-4, IL-17 or IL-10, etc, to more fully define Th lineage), 3) lack of HLA-phenotyping of subjects and 4) mixed inclusion (and analysis) of both treated and untreated samples. The rationale for such a short incubation period to assess stimulation index and such a long incubation period to detect cytokine production is not clearly explained in the manuscript.
- It would be informative to include time interval between blood sampling and clinical attacks
- Table 1 would be improved by including a row with information on additional immunomodulatory treatments received by subjects (e.g. azathioprine, methotrexate, etc.)
- It would be informative to show or state whether there were any differences in stimulation index comparing untreated (i.e. no corticosteroids or other immunomodulatory treatment exposure) versus treated samples
- Presentation of data would be enhanced by including representative plots/histograms of the flow cytometric analysis of intracellular cytokines
Author Response
Weaknesses of the study are
â–ˆ1) the lack of a measure of proliferation,
(Our response) We agree that it is a limitation of the present study and stated it in the 7th paragraph of Discussion (The present study has several limitations. …) of our original manuscript.
â–ˆ2) limited assessment of intracellular cytokines (i.e. lacks assessment of IL-4, IL-17 or IL-10, etc, to more fully define Th lineage),
(Our response) Thank you for the comment. We added it in Discussion (7th paragraph, The present study has several limitations. …).
â–ˆ3) lack of HLA-phenotyping of subjects.
(Our response) We added it in Discussion (7th paragraph, The present study has several limitations. …).
Line 283-285
“Moreover, limited evaluation of intracellular cytokines (IL-4, IL-17, IL-10, etc) and lack of HLA-phenotyping are also limitations of this study. Adding those data would be useful to further clarify T cell response to MOG peptides in MOGAD.”
â–ˆ4) mixed inclusion (and analysis) of both treated and untreated samples.
(Our response) We thank the reviewer for this valuable comment. We compared treated and untreated patients, but no significant difference of CD69 expression was observed. We added it in Results.
Line 98-99
“When comparing the treated and untreated MOGAD patients, no difference was observed in the expression of CD69 on T cells between the two groups.”
â–ˆThe rationale for such a short incubation (4 hours) period to assess stimulation index
(Our response) We decided on the incubation period based on previous reports (Matsuya et al. Int Immunol 2011 [Ref 9], Cibrian et al. Eur J Immunol 2017). Since CD69 expression can be detected as early as 2-3 hours after antigen stimulation, and 4 hours is considered appropriate to assess SI. We cited Ref 9 in our original manuscript.
â–ˆsuch a long incubation period to detect cytokine production (7 days) is not clearly explained in the manuscript.
(Our response) We adopted the incubation time (7 days) based on a previous report on T cell response to MOG and AQP4 peptides (Hofer et al. Front Immunol 2020 [19]). We cited the reference in our revised manuscript.
Line 323
“…concentration of 10 μg/mL for seven days [19].”
â–ˆIt would be informative to include time interval between blood sampling and clinical attacks.
(Our response) We added the time interval in Table 1.
â–ˆTable 1 would be improved by including a row with information on additional immunomodulatory treatments received by subjects (e.g. azathioprine, methotrexate, etc.)
(Our response) We added immunomodulatory treatments in Table 1.
â–ˆIt would be informative to show or state whether there were any differences in stimulation index comparing untreated (i.e. no corticosteroids or other immunomodulatory treatment exposure) versus treated samples
(Our response) We compared treated and untreated patients, but no significant difference of CD69 expression was observed. We added it in Results.
Line 98-99
“When comparing the treated and untreated MOGAD patients, no difference was observed in the expression of CD69 on T cells between the two groups.”
â–ˆPresentation of data would be enhanced by including representative plots/histograms of the flow cytometric analysis of intracellular cytokines
(Our response) We add the representative plots in a patient with MOGAD (43 years-old male, treated with PSL and AZA) in Figure 2A.
Reviewer 2 Report
Comments and Suggestions for Authors
The group of Ono et al. investigated the CD4-positive T cell response to MOG peptides in MOG-anti-
body-associated disease. The authors characterised the T cells on the basis of the cell surface CD69 and the intracellular production of granulocyte-macrophage colony-stimulating factor (GM-CSF) and interferon-γ. The present study showed a subtle CD4-positive T cell activation triggered by some MOG peptides alone in patients with MOGAD. I have the following comments: What is the connection between MOGAD and NMOSD? Why were both included in the study? Does this mean that MOGAD is a collection of different diseases with a very different picture? What is the distribution/frequency of the different forms of expression? Overall, MOGAD shows a different patient picture. So how comparable are the patients? Why were these three patients selected under point 2.4? What is the essence of the patients and the study? What conclusions can be drawn from this? For example, with regard to the treatment of the patients?
Author Response
I have the following comments:
â–ˆWhat is the connection between MOGAD and NMOSD? Why were both included in the study?
(Our response) Both MOGAD and NMOSD are autoantibody-associated inflammatory central nervous system diseases, and these two patients groups were compared in a previous study (Hofer et al. Front Immunol 2020 [19]).
â–ˆDoes this mean that MOGAD is a collection of different diseases with a very different picture? What is the distribution/frequency of the different forms of expression? Overall, MOGAD shows a different patient picture. So how comparable are the patients?
(Our response) The clinical phenotypes of MOGAD in our study were as follows; ADEM (n = 7), optic neuritis (n = 13), myelitis (n = 7), brainstem encephalitis (n = 5), cerebral cortical encephalitis (n = 2), and multiple brain lesions without encephalopathy (n = 7). None of these cases were diagnosed as seronegative NMOSD according to the international consensus diagnostic criteria for NMOSD. It is interesting to analyze the data based on the clinical phenotypes, but we could not do it due to the small number of cases.
â–ˆWhy were these three patients selected under point 2.4? What is the essence of the patients and the study? What conclusions can be drawn from this? For example, with regard to the treatment of the patients?
(Our response) These three cases were the ones that we could analyze the data both before and after acute treatment. After treatment, there was a tendency for SI to decrease compared to before treatment. As a result, CD69 expression seems to be associated with relapse to some extent although statistical analysis was not done due to the small number of cases.